DOI: 10.1038/s41467-018-05714-3　　**OPEN**

# Autosomal genetic variation is associated with DNA methylation in regions variably escaping X-chromosome inactivation

René Luijk[1], Haoyu Wu[2], Cavin K Ward-Caviness[3], Eilis Hannon[4], Elena Carnero-Montoro[5,22], Josine L. Min[6,7], Pooja Mandaviya[8,9], Martina Müller-Nurasyid [10,11,12], Hailiang Mei[2], Silvere M. van der Maarel[2], BIOS Consortium[#], Caroline Relton[6], Jonathan Mill[4], Melanie Waldenberger[3,13], Jordana T. Bell[5], Rick Jansen [14], Alexandra Zhernakova[15], Lude Franke [15], Peter A.C. 't Hoen [2], Dorret I. Boomsma[16], Cornelia M. van Duijn[17], Marleen M.J. van Greevenbroek[18,19], Jan H. Veldink[20], Cisca Wijmenga[15], Joyce van Meurs[8], Lucia Daxinger[2], P. Eline Slagboom[1], Erik W. van Zwet[21] & Bastiaan T. Heijmans [1]

X-chromosome inactivation (XCI), i.e., the inactivation of one of the female X chromosomes, restores equal expression of X-chromosomal genes between females and males. However, ~10% of genes show variable degrees of escape from XCI between females, although little is known about the causes of variable XCI. Using a discovery data-set of 1867 females and 1398 males and a replication sample of 3351 females, we show that genetic variation at three autosomal loci is associated with female-specific changes in X-chromosome methylation. Through *cis*-eQTL expression analysis, we map these loci to the genes *SMCHD1/METTL4*, *TRIM6/HBG2*, and *ZSCAN9*. Low-expression alleles of the loci are predominantly associated with mild hypomethylation of CpG islands near genes known to variably escape XCI, implicating the autosomal genes in variable XCI. Together, these results suggest a genetic basis for variable escape from XCI and highlight the potential of a population genomics approach to identify genes involved in XCI.

[1] Molecular Epidemiology, Department of Biomedical Data Sciences, Leiden University Medical Center, Leiden 2333 ZC, The Netherlands. [2] Department of Human Genetics, Leiden University Medical Center, Leiden 2333 ZC, The Netherlands. [3] Institute of Epidemiology II, Helmholtz Zentrum München, Neuherberg 85764 Oberschleißheim, Germany. [4] University of Exeter Medical School, Exeter EX4 4QD, UK. [5] Department of Twin Research & Genetic Epidemiology, King's College London, London SE1 7EH, UK. [6] MRC Integrative Epidemiology Unit, University of Bristol, Bristol BS8 1TH, UK. [7] Bristol Medical School, University of Bristol, Bristol BS8 1UD, UK. [8] Department of Internal Medicine, Erasmus University Medical Center, Rotterdam 3015 CE, The Netherlands. [9] Department of Clinical Chemistry, Erasmus University Medical Center, Rotterdam 3015 CE, The Netherlands. [10] DZHK (German Centre for Cardiovascular Research), partner site: Munich Heart Alliance, Munich 80802, Germany. [11] Institute of Genetic Epidemiology, Helmholtz Zentrum München - German Research Center for Environmental Health, Neuherberg D-85764, Germany. [12] Department of Medicine I, University Hospital Munich, Ludwig-Maximilians-University, Munich 80336, Germany. [13] Research Unit of Molecular Epidemiology, Helmholtz Zentrum München, Neuherberg D-85764, Germany. [14] Department of Psychiatry, VU University Medical Center, Neuroscience Campus Amsterdam, Amsterdam 1081 HV, The Netherlands. [15] Department of Genetics, University of Groningen, University Medical Centre Groningen, Groningen 9713 AV, The Netherlands. [16] Department of Biological Psychology, Vrije Universiteit Amsterdam, Neuroscience Campus Amsterdam, Amsterdam 1081 TB, The Netherlands. [17] Department of Epidemiology, Genetic Epidemiology Unit, ErasmusMC, Rotterdam 3015 GE, The Netherlands. [18] Department of Internal Medicine, Maastricht University Medical Center, Maastricht 6211 LK, The Netherlands. [19] School for Cardiovascular Diseases (CARIM), Maastricht University Medical Center, Maastricht 6229 ER, The Netherlands. [20] Department of Neurology, Brain Center Rudolf Magnus, University Medical Center Utrecht, Utrecht 3584 CG, The Netherlands. [21] Medical Statistics, Department of Biomedical Data Sciences, Leiden University Medical Center, Leiden 2333 ZC, The Netherlands. [22] Present address: Pfizer - University of Granada - Andalusian Government Center for Genomics and Oncological Research (GENYO), Granada 18016, Spain. Correspondence and requests for materials should be addressed to . B.T.H. (email: b.t.heijmans@lumc.nl). A full list of consortium members appears at the end of the paper.

To achieve dosage equivalency between male and female mammals, one of two X-chromosomes is silenced early in female embryonic development resulting in one inactive (Xi) and one active (Xa) copy of the X-chromosome[1]. While the Xi-linked gene *XIST* is crucial for the initiation of X-chromosome inactivation (XCI), autosomal genes appear to be critically involved in XCI establishment and maintenance[2]. An abundance of repressive histone marks[3–5] and DNA methylation[6,7] throughout XCI on Xi is in line with a prominent role of epigenetic regulation in both phases. However, the Xi is not completely inactivated. With an estimated 15% of X-chromosomal genes consistently escaping XCI, and an additional 10% escaping XCI to varying degrees[8,9], escape from XCI is fairly common in humans[10–12], much more so than in mice[13]. Genes escaping XCI are characterized by distinct epigenetic states[14] and are thought to be associated with adverse outcomes, including mental impairment[12–14].

In the mouse, an example of an autosomal gene involved in XCI is *Smchd1*. *Smchd1* is an epigenetic repressor and plays a critical role in the DNA methylation maintenance of XCI in mice[15,16]. However, in humans, in-depth knowledge on the role of autosomal genes in XCI maintenance is lacking, despite earlier in vitro[17] efforts. Furthermore, the mechanisms underlying variable XCI, a common feature of human XCI[8,9], are unknown.

Here, we report on the identification of four autosomal loci associated with female-specific changes in X-chromosome DNA methylation using a discovery set of 1867 females and 1398 males, and replication of three of these loci in an independent replication set consisting of 3351 female samples. The replicated loci map to the genes *SMCHD1/METTL4*, *TRIM6/HBG2*, and *ZSCAN9* through eQTL analysis. All three preferentially influenced the methylation of CpGs located in CpG islands (CGI) near genes known to variably escape XCI between individuals, providing evidence for a genetic basis of this phenomenon.

## Results

### Identifying female-specific genetic effects on X-methylation.
To identify genetic variants involved in XCI, we employed a global test approach[18] to evaluate the association of 7,545,443 autosomal genetic variants with DNA methylation at any of 10,286X-chromosomal CpGs measured in whole blood of 1867 females (Supplementary Data 1) using the Illumina 450k array (see 'Methods'). The analysis was corrected for covariates, including cell counts, age, and batch effects. We identified 4504 individual variants representing 48 independent loci associated with X-chromosomal methylation in females (Wald $P < 5 \times 10^{-8}$, Fig. 1 and Supplementary Fig. 1), each defined by the most strongly associated variant (as reflected by the lowest $P$-value), termed the sentinel variant. Of the 48 sentinel variants corresponding to these 48 loci, 44 were also associated with X-chromosomal methylation in males ($N = 1398$, Supplementary Data 1, Supplementary Data 2; Wald $P < 1.1 \times 10^{-6}$) indicating that the associations were unrelated to XCI. The four remaining variants did not show any indication for an effect in males (Wald $P > 0.19$) while they did show strong, widespread, and consistent same-direction effects across the X-chromosome in females (Supplementary Data 2, Supplementary Data 3, Supplementary Fig. 2). Formally testing for a genotype by sex interaction revealed significant interaction effects for three of the four variants. The rs140837774, rs139916287, and rs1736891 variants with evidence for an interaction ($P_{interaction} < 5.9 \times 10^{-4}$) mapped to the *SMCHD1/METTL4*, *TRIM6/HBG2*, and *ZSCAN9* loci, respectively, (see 'Methods'). The remaining variant rs73937272 ($P_{interaction} = 0.88$) mapped near the *ZNF616* gene. Finally, we evaluated whether the

effect of the autosomal loci was influenced by genetic variation on X, but this did not change the results (Supplementary Data 4, see 'Methods').

To establish the validity and stability of the analyses, we first investigated whether any of the associations were due to confounding by cellular heterogeneity. Therefore, we directly tested for an association between the four identified sentinel variants and the observed red and white blood cell counts. This did not result in any significant association (Supplementary Fig. 3). Furthermore, we determined that none of the four identified variants are among the variants known to affect blood composition[19,20]. Vice versa, genetic variants known to affect blood cell counts also did not show an association with X-chromosomal methylation in our data (Supplementary Fig. 4, Supplementary Data 5). Re-testing the effects of the four sentinel variants while adjusting for nearby blood composition-associated SNPs (<1 Mb) did not influence the results (Supplementary Data 6, see 'Methods'). Second, we addressed unknown confounding by including latent factors as covariates in our models, estimated in the methylation data using software for estimation and adjustment of unknown confounders in high-dimensional data[21] (see 'Methods'). Re-testing the four sentinel variants without these latent factors did not change the results (Supplementary Data 6). We conclude that the effects of the four variants identified in the discovery data are stable and not confounded by cellular heterogeneity or other, unknown, factors.

Finally, we tested the four sentinel variants in an additional 3351 unrelated female samples (see 'Methods' and Supplementary Data 7), and successfully replicated the rs140837774, rs139916287, and rs1736891 variants (Bonferroni corrected, $P_{adj} = 0.0096$, $P_{adj} = 2.4 \times 10^{-4}$, and $P_{adj} = 2.2 \times 10^{-3}$, respectively). The rs73937272 variant ($P_{adj} = 1$), which also lacked a sex-genotype interaction effect in the discovery set, was not replicated. In further analyses, we focussed on the three replicated loci.

### Exploration of genetic loci affecting X-methylation.
The sentinel variant rs140837774 is an AATTG insertion/deletion variant (MAF = 0.49) on chromosome 18, located in intron 26 of *SMCHD1* (Supplementary Fig. 2), a gene known to be critically involved in XCI in mice[15,22–25]. In addition, *SMCHD1* mutations affect the methylation levels of the *D4Z4* repeat in humans, playing an important role in facioscapulohumeral dystrophy 2 (FSHD2)[26]. To link rs140837774 to a nearby gene we performed a *cis*-eQTL analysis using RNA-seq data from the 1867 females in the discovery set of our study (250 Kb upstream and downstream of the sentinel variant, Supplementary Data 8). We found that the deletion was strongly associated with decreased *SMCHD1* expression (Fisher's $P = 1.8 \times 10^{-10}$, regression coefficient = −0.13) and increased expression of the methyltransferase *METTL4*, albeit weaker (Wald $P = 4.9 \times 10^{-4}$, regression coefficient = 0.04). *METTL4* is a highly conserved gene[27,28], involved in the mRNA modification N6-methyladenosine (reviewed in ref. [29]), which plays an important role in epigenetic regulation in mammals[30].

The *SMCHD1/METTL4* variant was associated with altered methylation levels of 57X-chromosomal CpGs in females (FDR < 0.05, Fig. 2a, Supplementary Data 9). The deletion (the low *SMCHD1* expression allele) was associated with hypomethylation at 56 of those X-chromosomal CpGs (98.2%, binomial $P = 8.5 \times 10^{-13}$ Fig. 2a), consistent with X-hypomethylation in female mice deficient for *SMCHD1*[25]. The mean effect size was 1% per rare allele (ranging from 0.27 to 2.34%, Supplementary Fig. 5), with the mean methylation values per CpG ranging from 2.6 to 55% (average methylation at these 56 CpGs is 23.6%).

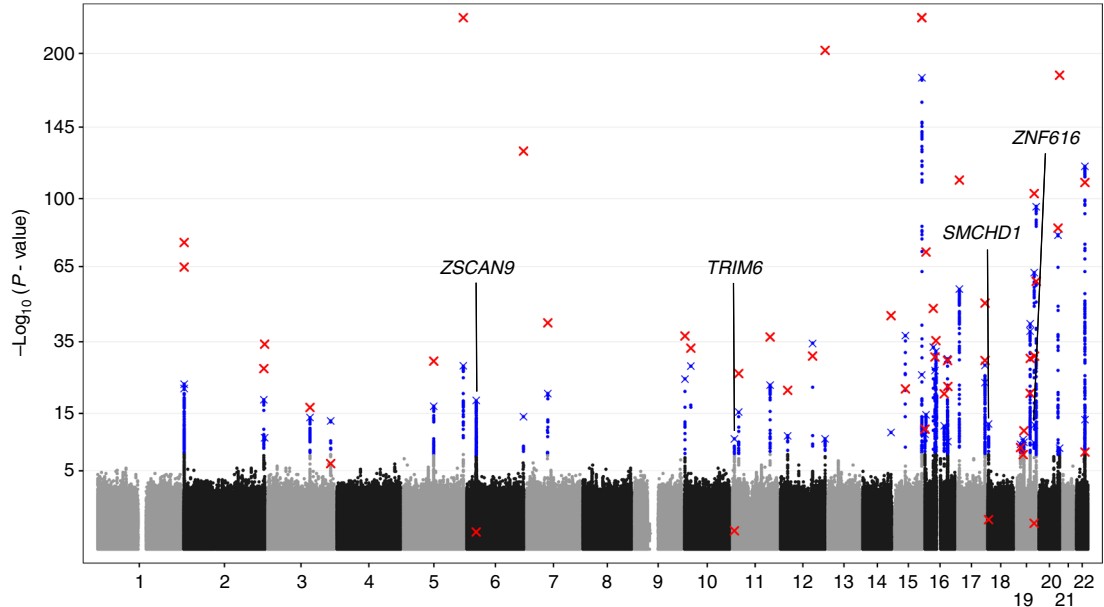

**Fig. 1** Manhattan plot showing all tested autosomal SNPs for an overall effect on X-chromosomal methylation in females. Significant associations are depicted in blue (Wald $P < 5 \times 10^{-8}$). The sentinel variant per independent locus is indicated with a blue cross. Testing the effects of these 48 sentinel variants in males, we found 44 replicated in males (Wald $P < 1.1 \times 10^{-6}$, red cross), whereas the other 4 loci were female-specific, as they clearly lacked an effect in males (Wald $P > 0.19$)

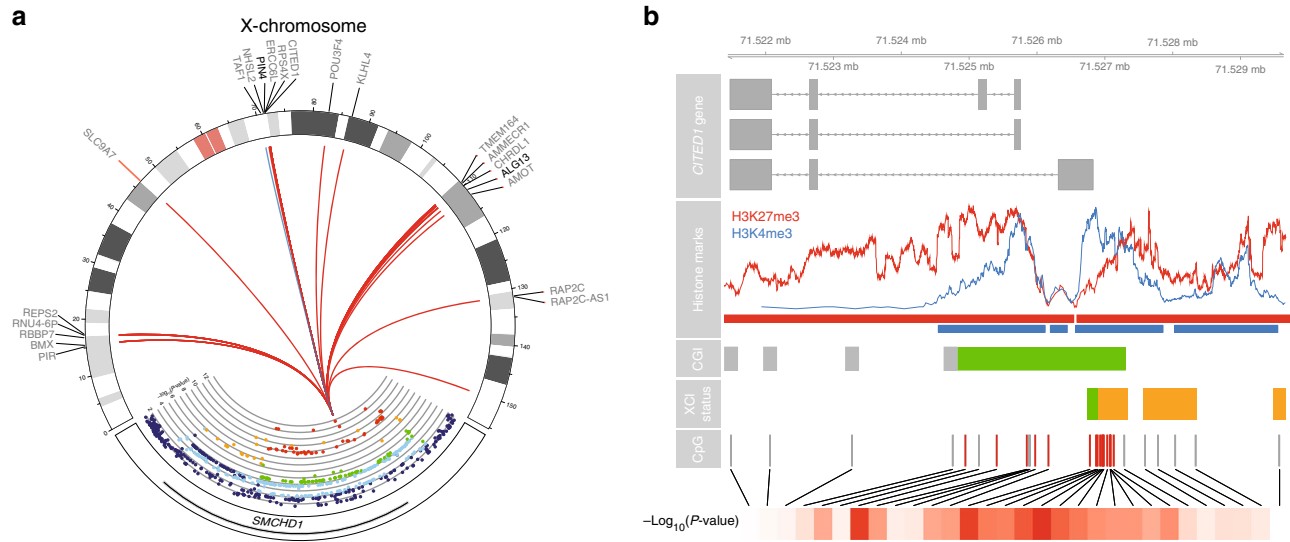

**Fig. 2** The *SMCHD1/METTL4* locus associates with DNA methylation at X-chromosomal and autosomal regions. **a** Plot showing the *SMCHD1/METTL4* locus and the effects it has on the X-chromosome. The colors in the *SMCHD1/METTL4* locus indicate LD (red: $R^2 \geq 0.8$; orange: $0.6 \leq R^2 < 0.8$; green: $0.4 \leq R^2 < 0.6$; light blue: $0.2 \leq R^2 < 0.4$; dark blue: $R^2 \ 0.6 \leq 0.2$). The y-axis shows the $-\log_{10}(P\text{-value})$ of the association with overall X-chromosomal methylation. The line colors in the Circos plot indicate the direction of the effect (red: hypomethylation, blue: hypermethylation). The plot of the *SMCHD1/METTL4* locus mostly shows moderate to high LD and covers most of the *SMCHD1* gene. The deletion of rs140837774 is associated with both downregulation of *SMCHD1* and upregulation of *METTL4* (see main text). Its effects on X-chromosomal methylation are abundant and consistent: the deletion of rs140837774 is associated with hypomethylation at 98.2% of all associated CpGs (56 CpGs, red lines, mean effect size 1% per allele). Hypomethylation of CpGs near two genes known to escape XCI to varying degrees[12] (*PIN4* and *ALG13*, shown in bold) is associated with increased expression of these two genes. **b** Example of CpG island (CGI) in the *CITED1* gene (first row) associated with the *SMCHD1/METTL4* locus. The CpGs associated with the *SMCHD1/METTL4* locus (fifth row, indicated by red lines) are overrepresented in regions characterized by both active (blue) and repressive (red) histone marks (second row, red and blue bars, two-fold enrichment, Fisher's $P = 1.5 \times 10^{-12}$; 6.9-fold enrichment, Fisher's $P = 1.5 \times 10^{-12}$), are often located in CpG islands (third row, green bar, 11.3-fold enrichment, Fisher's $P = 2.5 \times 10^{-14}$), and regions known to variably escape X-chromosome inactivation[10] (fourth row, orange bars, 21.4-fold enrichment, Fisher's $P = 3.7 \times 10^{-18}$). The bottom row indicates the strength of the associations in terms of $-\log_{10}(P\text{-value})$ (dark red indicates strong associations)

Compared to all X-chromosomal CpGs in our data, the associated X-chromosomal CpGs were strongly overrepresented in CGI (50 out of 57 CpGs, 11.3-fold enrichment, binomial $P = 2.5 \times 10^{-14}$, Fig. 2b), in line with *SMCHD1*'s role in X-chromosomal CGI methylation[24]. Data on chromatin marks in blood[31] (see 'Methods') revealed a strong overrepresentation of the associated X-chromosomal CpGs in regions bivalently marked by the active histone mark H3K4me3 (47 CpGs, 8.2-fold enrichment, Fisher's $P = 1.5 \times 10^{-12}$), and the repressive mark H3K27me3 (38 CpGs, 6.9-fold enrichment, Fisher's $P = 1.5 \times 10^{-12}$), as compared to all X-chromosomal CpGs in our data. In agreement with this, we observed a 16.9-fold enrichment for CpGs overlapping bivalent/poised transcription start sites (TSSs) (35/57 CpGs, Fisher's $P = 4.3 \times 10^{-23}$) using predicted chromatin segmentations[31], possibly reflecting the mixed signals from both the active and inactive X chromosomes underlying these chromatin segmentations. Strikingly, annotation by the degree of escape for 489 TSSs in 27 different tissues, and specifically whole blood[10] (see 'Methods'), revealed a strong overrepresentation of CpGs located near TSSs variably escaping XCI (22 CpGs, 21.4-fold enrichment, Fisher's $P = 3.7 \times 10^{-18}$, Fig. 2b). Only a modest enrichment for associated CpGs in fully escaping XCI regions (15 CpGs, 4.2-fold enrichment, Fisher's $P = 4.5 \times 10^{-5}$) and an underrepresentation of associated CpGs in inactivated regions (7 CpGs, 28.6-fold depletion, Fisher's $P = 2.2 \times 10^{-23}$) was observed.

Further supporting a link with variable XCI, we observed that X-chromosomal CpGs were associated with differential expression of the nearby genes (<250 Kb, see 'Methods') *ALG13* and *PIN4* (see 'Methods', Supplementary Data 10) both known to variably escape XCI[12]. While a strong eQTL effect and a clear biologically relevant link with XCI mainly implies *SMCHD1* in X-chromosomal hypomethylation (insertion of rs140837774), an eQTL effect for *METTL4*, although slightly weaker, leaves open a

possible role for *METTL4* in XCI, given its role in the mRNA modification N⁶-methyladenosine.

Using both female and male samples ($N = 3265$, Supplementary Fig. 6) to investigate associations of genetic variation at the *SMCHD1/METTL4* locus with autosomal methylation in trans (>5 Mb), we found that the *SMCHD1/METTL4* variant was associated with 20 CpGs mapping to the *HOXD10*, *HOXC10*, and *HOXC11* genes of the *HOXD* and *HOXC* clusters located on chromosomes 2 and 12, as well as to the large protocadherin beta (*PCDHβ*) and gamma (*PCDHγ*) clusters on chromosome 5 (FDR < 0.05, Supplementary Fig. 7, Supplementary Data 11), all known *SMCHD1* targets[25,32].

The second of the three sentinel variants, sentinel SNP rs139916287 (MAF = 0.07), is located in intron 4 of the *HBG2* gene on chromosome 11, in the β-globin locus (rs139916287, chromosome 11, Supplementary Fig. 2B). The rare allele of the sentinel variant was associated with decreased expression of both the *HBG2* and *TRIM6* genes (T allele; Wald $P = 5.3 \times 10^{-7}$, regression coefficient = −130.55; Wald $P = 9.8 \times 10^{-5}$, regression coefficient = −0.05; Supplementary Data 8), based on cis-eQTL mapping testing genes up to 250 kb upstream, and downstream of the sentinel variant (Supplementary Data 8). While *HBG2* showed higher expression levels and a stronger eQTL effect in our data, *TRIM6* has been shown to bind *XIST*[33], and contributes to the maintenance of pluripotency in mouse embryonic stem cells[34], making *TRIM6* a strong candidate for explaining our observations. Associating the *TRIM6/HBG2* variant with X-chromosomal methylation, we found 276 associated X-chromosomal CpG sites (FDR < 0.05, Fig. 3a, Supplementary Data 9). The rare allele (T allele) was associated with hypomethylation at 258 of those CpGs (93.5%, binomial $P = 6.3 \times 10^{-47}$), where mean effect size at these 258 CpGs is 1.6% per T allele, ranging from 0.15% to 4.25% (Supplementary Fig. 5).

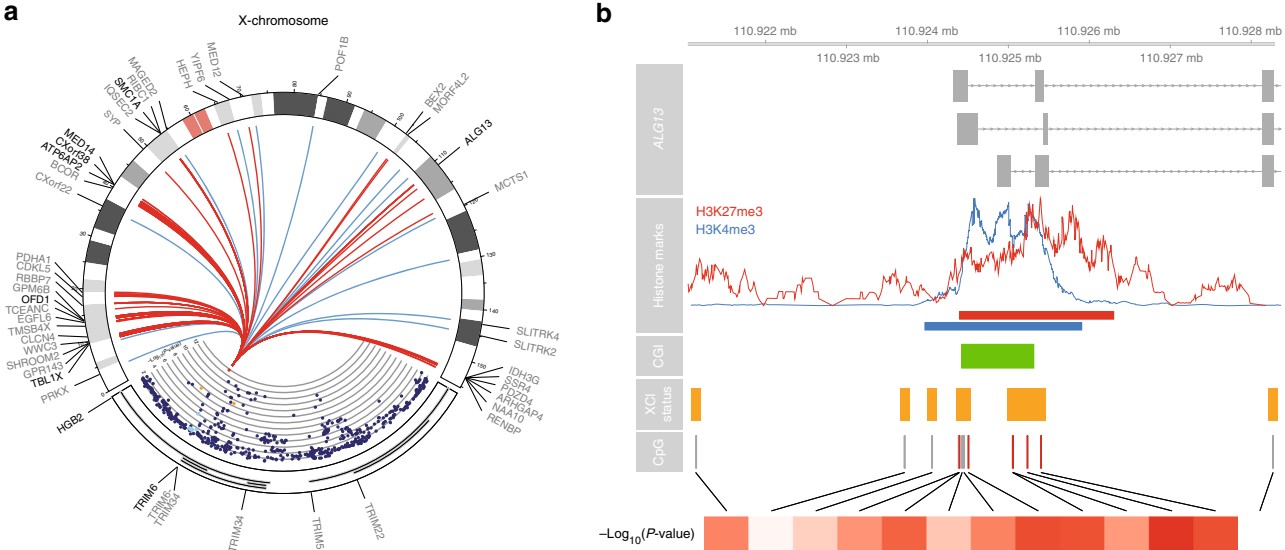

**Fig. 3** The *TRIM6/HBG2* locus is associated with DNA methylation at X-chromosomal regions. **a** Plot showing the *TRIM6/HBG2* locus and the widespread effects it has on the X-chromosome. The colors in the *TRIM6/HBG2* locus indicate LD (red: $R^2 \geq 0.8$; orange: $0.6 \leq R^2 < 0.8$; green: $0.4 \leq R^2 < 0.6$; light blue: $0.2 \leq R^2 < 0.4$; dark blue: $R^2$ $0.6 \leq 0.2$). The y-axis shows the $-\log_{10}(P\text{-value})$ of the association with overall X-chromosomal methylation. The line colors in the Circos plot indicate the direction of the effect (red: hypomethylation, blue: hypermethylation). The T allele of its sentinel variant rs139916287 is associated with upregulation of *HBG2*, downregulation of *TRIM6* (both shown in bold), and hypomethylation at 258 of the 276 associated CpGs (93.5%, red lines, mean effect size 1.6% per allele). X-chromosomal genes whose expression levels were associated with methylation levels of nearby CpGs are shown in bold. **b** Example of CpG island (CGI) in the *ALG13* gene associated with the *TRIM6/HBG2* locus. The enrichments of CpGs in certain genomic regions are similar to those found for the *SMCHD1/METTL4* locus. Most notably, the associated CpGs are also overrepresented in regions known to variably escape X-chromosome inactivation[10] (fourth row, orange bars, 8.8-fold enrichment, Fisher's $P = 2.1 \times 10^{-20}$)

Similar to the *SMCHD1/METTL4* variant, associated CpGs were overrepresented in CGIs (199 CpGs, 4.2-fold enrichment, Fisher's $P = 1.6 \times 10^{-29}$), and enriched in regions characterized by H3K27me3 (208 CpGs, 10.5-fold enrichment, Fisher's $P = 3.5 \times 10^{-77}$) and H3K4me3[31] (217 CpGs, 6.4-fold enrichment, Fisher's $P = 5.5 \times 10^{-46}$, Fig. 3b). The associated CpGs were again strongly overrepresented in genomic regions variably escaping XCI in an external set of whole blood samples[10] (see 'Methods', 39 CpGs, 8.8-fold enrichment, Fisher's $P = 2.1 \times 10^{-20}$), to a lesser extent present in regions consistently escaping XCI (61 CpGs, 6.8-fold enrichment, Fisher's $P = 2.2 \times 10^{-23}$), and underrepresented in repressed regions (39 CpGs, 15.2-fold depletion, Fisher's $P = 1.9 \times 10^{-50}$).

In addition, many genes known to variably escape XCI[12], annotated to CpGs associated with *TRIM6/HBG2* genetic variation (*ALG13*, *ATP6AP2*, *CXorf38*, *MED14*, *SMC1A*, *TBL1X*, Supplementary Data 10). Similar to *SMCHD1/METTL4*, these results suggest a role for the *TRIM6/HBG2* locus in variable escape from XCI.

The sentinel SNP of the third identified locus (rs1736891, MAF = 0.38) was associated with the expression of several nearby genes annotated as zinc fingers (Supplementary Data 8), but most strongly with downregulation of the expression of the nearby transcription factor[35] *ZSCAN9* gene located on chromosome 6, based on cis-eQTL mapping in our own data (Wald $P = 2.5 \times 10^{-49}$, Supplementary Data 8). The sentinel SNP was significantly associated with 19 X-chromosomal CpGs in females (FDR < 0.05, Supplementary Fig. 8, Supplementary Data 9), all located in the same CGI: the high-expression A allele of rs1736891 was associated with mild hypomethylation of all 19 CpGs (Fisher's $P = 3.6 \times 10^{-5}$, mean effect size 1.3% per allele, Supplementary Fig. 5). There was an overlap of in the CpGs associated with the sentinel variants of the *ZSCAN9* and the *SMCHD1/METTL4* locus, although the two loci are located on different chromosomes (chromosomes 6 and 18, respectively, Supplementary Fig. 2). These associations were statistically independent from each other (i.e., additive), as all identified loci were identified using conditional analyses (see 'Methods'). Specifically, 17 out of 19 CpGs (89.5%) were also targeted by the *SMCHD1/METTL4* locus, and all 19 CpGs show consistent opposite effects for both loci (Supplementary Fig. 9). Similar to the *SMCHD1/METTL4* locus, the *ZSCAN9* locus also associated with autosomal DNA methylation in trans (>5 Mb). However, none of the autosomal CpGs overlapped between the two loci (Supplementary Data 11).

## Discussion

Here, we identify three autosomal genetic loci with female-specific effects on X-chromosomal methylation in humans (*SMCHD1/METTL4*, *TRIM6/HBG2*, and *ZSCAN9*), all of which were associated with altered expression of autosomal genes in cis. Furthermore, all three loci were consistently associated with mild hypomethylation of CpGs overrepresented in CGI of X-chromosomal regions known to variably escape XCI in whole blood[10,12]. The former finding extended to 26 other tissues[10], suggesting a cross-tissue genetic basis for variable escape from XCI. We observed a striking underrepresentation of affected CpGs in fully inactivated CGIs. This may be due to the tightly regulated nature of these regions. Methylation of these CpGs may be impervious to the impact of autosomal genetic variation or effects may be substantially weaker requiring much larger data sets to detect them.

While most of the previous work on XCI was done using mouse models and established a critical role for *SMCHD1* in XCI[15,23,25], we here confirm the role of the *SMCHD1/METTL4* locus in XCI in humans and highlight its impact on variable escape from XCI. This phenomenon has not been previously described in mice, perhaps due to the lack of genetic variability in the often inbred mice, leading to less (variable) escape from XCI than occurs in humans[8,9]. We also observed associations of the *SMCHD1/METTL4* locus with known autosomal SMCHD1 targets[25,32], most notably the protocadherin clusters[36]. Interestingly, similar to the X-chromosome, the expression of the clustered protocadherin genes is stochastic and mono-allelic[37], suggesting a common mechanism.

In addition to the *SMCHD1/METTL4* locus, our results indicated a role for the *TRIM6/HBG2* locus in XCI. *TRIM6* is a strong candidate to influence female X-chromosome methylation because it was reported to bind XIST[33] and is involved in *MYC* and *NANOG* regulation[34]. Similarly, our data suggest a role for the *ZSCAN9* locus in variable escape from XCI, as it affects a single CGI that is also targeted by the *SMCHD1/METTL4* locus. While this does suggest a role for the two loci in the same pathway, the effects on the X-chromosome were statistically independent.

Given the biological consistency of the findings presented here, and the replication thereof in an independent set of samples, our data support a role of autosomal genetic variants in regulating Xi methylation in particular at variably escaping regions. However, to definitely demonstrate causality, unequivocally identify the responsible genes, and provide precise insight into the exact underlying mechanisms, in vitro experiments are needed. Importantly, a population genomics approach, like ours, will reveal effects on both XCI establishment and maintenance, which occur during different developmental stages and may involve different molecular pathways. At this point, the exact role of the *SMCHD1/METTL4*, *TRIM6/HBG2*, and *ZSCAN9* loci during these processes remain to be determined. Therefore, it will be crucial to design experiments that can discriminate between an effect during the establishment and maintenance phases.

In conclusion, variable escape from XCI in humans has a genetic basis and we identified three autosomal loci, one previous implicated in XCI in mice and two new loci, that influence regions that are susceptible to variable escape from XCI by controlling X-chromosomal DNA methylation or correlated epigenetic marks.

## Methods

**Discovery cohorts**. The Biobank-based Integrative Omics Study (BIOS) Consortium comprises six Dutch biobanks: Cohort on Diabetes and Atherosclerosis Maastricht (CODAM)[38], LifeLines-DEEP (LLD)[39], Leiden Longevity Study (LLS)[40], Netherlands Twin Registry (NTR)[41], Rotterdam Study (RS)[42], Prospective ALS Study Netherlands (PAN)[43]. Institutional review boards of all cohorts approved this study (CODAM, Medical Ethical Committee of the Maastricht University; LL, Ethics committee of the University Medical Centre Groningen; LLS, Ethical committee of the Leiden University Medical Center; PAN, Institutional review board of the University Medical Centre Utrecht; NTR, Central Ethics Committee on Research Involving Human Subjects of the VU University Medical Centre; RS, Institutional review board (Medical Ethics Committee) of the Erasmus Medical Center). In addition, informed consent was provided by all participants. The data that were analyzed in this study came from 3265 unrelated individuals (Supplementary Data 1). Genotype data, DNA methylation data, and gene expression data were measured in whole blood for all samples. In addition, sex, age, measured cell counts (lymphocytes, neutrophils, monocytes, eosinophils, basophils, and red blood cell counts), and information on technical batches were obtained from the contributing cohorts. The Human Genotyping facility (HugeF, Erasmus MC, Rotterdam, The Netherlands, http://www.glimdna.org) generated the methylation and RNA-sequencing data and supplied information on technical batches.

Genotype data were generated within each cohort. Details on the genotyping and quality control methods have previously been detailed elsewhere (LLD: Tigchelaar et al.;[39] LLS: Deelen et al.[44], 2014; NTR: Lin et al.;[45] RS: Hofman et al.;[42] PAN: Huisman et al.[43]).

For each cohort, the genotype data were harmonized towards the Genome of the Netherlands[46] (GoNL) using Genotype Harmonizer[47] and subsequently imputed per cohort using Impute2[48] and the GoNL reference panel (v5)[46]. We removed SNPs with an imputation info-score below 0.5, a HWE P-value < $10^{-4}$, a call rate below 95% or a minor allele frequency smaller than 0.01. These imputation

and filtering steps resulted in 7,545,443 SNPs that passed quality control in each of the datasets.

A detailed description regarding generation and processing of the gene expression data can be found elsewhere[49]. Briefly, total whole blood was processed using Illumina's Truseq version 2 library preparation kit. Illumina's Hiseq2000 was used for paired-end sequencing. Lastly, CASAVA was used to create read sets per sample, applying Illumina's Chastity Filter. Data generation was done by the Human Genotyping facility (HugeF, ErasmusMC, The Netherlands, see URLs). QC was performed using FastQC (v0.10.1), cutadapt (v1.1)[50], and Sickle (v1.2)[51], after which the sequencing reads were mapped to human genome (HG19) using STAR (v2.3.0e)[52].

All common GoNL SNPs (MAF > 0.01, http://www.nlgenome.nl/?page_id=9) were masked with N to avoid reference mapping bias. Read pairs used were those with fewer than nine mismatches, and mapping to fewer than six positions.

Gene expression quantification was determined using base counts (for a detailed description, see ref.[49]). The gene definitions used for quantification were based on Ensembl version 71. For data analysis, we used reads per kilobase per million mapped reads (RPKM), and only used protein coding genes with sufficient expression levels (median RPKM > 1), resulting in a set of 10,781 genes. To limit the influence of any outliers still present in the data, the data were transformed using a rank-based inverse normal transformation within each cohort.

The Zymo EZ DNA methylation kit (Zymo Research, Irvine, CA, USA) was used to bisulfite-convert 500 ng of genomic DNA, and 4 μl of bisulfite-converted DNA was measured on the Illumina HumanMethylation450 array using the manufacturer's protocol (Illumina, San Diego, CA, USA). Preprocessing and normalization of the data were done as described earlier[53]. Removal of ambiguously mapped probes or probes containing known common genetic variants[54] were removed, followed by quality control (QC) using MethylAid's[55] default settings, investigating methylated and unmethylated signal intensities, bisulfite conversion, hybridization, and detection P-values. Filtering of individual beta-values was based on detection P-value (P < 0.01), number of beads available (≤2) or zero values for signal intensity. Normalization was done using Functional Normalization[56] as implemented in the minfi R package[57], using five principal components extracted using the control probes for normalization. All samples or probes with more than 5% of their values missing were removed, based on the QC performed using MethylAid. The final dataset consisted of 440,825 probes measured in 3265 samples. Lastly, similar to the RNA-sequencing data, the methylation data were also transformed using a rank-based inverse normal transformation within each cohort, to limit the influence of any remaining outliers while removing any systematic differences in mean methylation between cohorts.

**Replication cohorts.** Samples were drawn from the Avon Longitudinal Study of Parents and Children (ALSPAC) (Fraser et al. 2013; Boyd et al. 2013). Blood from 1022 mother–child pairs (children at three time points and their mothers at two time points) were selected for analysis as part of Accessible Resource for Integrative Epigenomic Studies (ARIES, http://www.ariesepigenomics.org.uk/) (Relton et al. 2015). Written informed consent has been obtained for all ALSPAC participants. Ethical approval for the study was obtained from the ALSPAC Ethics and Law Committee and the Local Research Ethics Committees. Genotyping and methylation measurements have been previously described[58,59].

The University College London case-control sample has been described elsewhere[60,61] but briefly comprises of unrelated ancestrally matched schizophrenia cases recruited from NHS mental health services and controls from the United Kingdom. All controls were screened to exclude the presence of an RDC defined mental disorder, alcohol dependence, or a family history thereof. Informed consent was obtained from all participating subjects, as well as UK National Health Service multicentre and local research ethics approval. Details of DNA methylation and genetic data generation, processing, quality control and normalisation can be found in the original EWAS manuscript[61].

The Aberdeen case-control sample has been described elsewhere[61,62] but briefly contains schizophrenia cases and controls who have self-identified as born in the British Isles (95% in Scotland). Controls were selected based on age, sex, and excluded if they presented with a mental disorder, or one of their first degree relatives, or if they used neuroleptic medication. All subjects gave informed consent. Different ethical committees (both local and multiregional) approved of the study. Details of DNA methylation and genetic data generation, processing, quality control and normalisation can be found in the original EWAS manuscript[61].

The KORA study (Cooperative health research in the Region of Augsburg) consists of independent population-based samples from the general population living in the region of Augsburg, Southern Germany. Written informed consent has been given by each participant and the study was approved by the local ethical committee. The dataset comprised individuals from the KORA F4 survey (all with genotyping and methylation data available) conducted from 2006–2008.

The TwinsUK cohort recruits monozygotic and dizygotic twins since its establishment in 1992[63], includes over 13,000 twin participants, and is a good representation of the general population in the UK and represents this population with respect to several phenotypes, including disease[64]. As such, it has been used in several epidemiological studies.

The Rotterdam Study (RS) is a prospective, population-based cohort study, investigating chronic diseases in the elderly[65]. The study consists of several cohorts

(RS-I, RS-II, RS-III), totalling 14,926 subjects from the Ommoord district in Rotterdam, The Netherlands. All participants gave informed consent. This study was approved by the Erasmus University Medical Center medical ethics committee.

**Discovering female-specific genetic effects on X-methylation.** To identify autosomal genetic variants influencing DNA methylation anywhere on the X-chromosome we applied a two-step approach[18] using 1867 female samples from the replication cohorts for which both genotype data and methylation data were available. We first fitted linear models to test for an association between each autosomal SNP $y_j$ and each of 10,286 X-chromosomal CpGs $x_i$ individually, correcting for known covariates $M$ (cell counts, cohort, age, technical batches—e.g., sample plate and array position) and unknown confounding by including latent factors $U$, estimated using cate[21], where the eigenvalue difference method implemented in cate suggested an optimal number of three latent factors:

$$y_j = \beta_{ij} x_i + \gamma M + \delta U \qquad (1)$$

For each autosomal genetic variant $i$, this approach yields 10,286 P-values $p_{ij}$. Next, we combined all 10,286 P-values corresponding to one genetic variant $i$ into one overall P-value $_{pi}$ using the Simes procedure[66], yielding 7,545,443 P-values $_{pi}$, one for each autosomal genetic variant tested. This overall P-value per SNP indicates if an autosomal SNP influences DNA methylation anywhere on the X-chromosome, reducing this analysis to a GWAS for X-chromosomal DNA methylation. SNPs with an overall P-value $< 5 \times 10^{-8}$ were deemed significantly associated with X-chromosomal DNA methylation.

To identify independent effects among the identified variants, we performed iterative conditional analyses. We re-ran the entire above procedure, correcting for the strongest associated sentinel variant, as determined by the lowest overall P-value.

$$y_j = \beta_{ij} x_i + \gamma M + \delta U + \beta_{\text{top SNP}} x_{\text{sentinel}} \qquad (2)$$

Having identified a new top SNP at the same genome-wide significance level of $P < 5 \times 10^{-8}$, we again re-did our analysis, now correcting for two top SNPs. We repeated this process until no new independent effects were identified, which was after 47 such iterations, thus yielding 48 sentinel variants, corresponding to 48 different loci.

Next, to establish the female-specificity of the identified loci on X-chromosomal methylation, we aimed to validate the 48 identified loci in 1398 males from the discovery cohorts for which the same genotype and methylation data were available. Any locus also having an effect in males would then mean that particular locus was not female specific. To do this, we tested the sentinel variant per locus found in females in the exact same way as we did in females, but also testing all SNPs within 1 Mb correlated to the sentinel variant ($R^2 \geq 0.8$ in males). A locus with any SNP having an overall P-value $_{pi} \leq 0.05$ in males was not considered to be female-specific, yielding four loci with four corresponding sentinel variants.

**Replicating female-specific genetic effects on X-methylation.** To replicate the four identified sentinel variants, we used an independent sample of 3351 females from five different replication cohorts (see section Description of replication cohorts), all having genotype and 450 k methylation data available. Similar to the discovery phase, each of four sentinel variants $x_i$ was associated with all X-chromosomal CpGs $y_j$ in each replication cohort $k$:

$$y_{jk} = \beta_{ijk} x_{ik} \qquad (3)$$

each yielding a test-statistic $t_{ijk}$. We then combined the test-statistics corresponding to each genetic variant $i$ and CpG $j$ between each cohort $j$ using Stouffer's weighted Z-method (discussed in ref.[67]), resulting in one overall Z-score $Z_{ij}$ for each variant-CpG pair $i,j$:

$$Z_{ij} = \frac{\sum_k w_k t_{ijk}}{\sqrt{\sum_k w_k^2}} \qquad (4)$$

where $w_k$ indicates the sample size for replication cohort $k$. Converting each overall Z-score $Z_{ij}$ to a P-value $p_{ij}$, we again used the Simes' procedure[66] to calculate one overall P-value $_{pi}$ per genetic variant $i$, representing the statistical evidence for an association with any X-chromosomal CpG in the replication cohorts.

**Local (cis) expression QTL mapping.** In order to map the identified sentinel variants associated with female-specific X-chromosomal methylation to nearby genes, we employed cis-eQTL mapping in the discovery cohorts, where we associated the genotypes of a genetic variant $i$ with the expression levels of genes $j$ in cis (<250 Kb). Similar to the trans-meQTL mapping for chromosome X, we corrected for known covariates $M$ (i.e., cell counts, cohort, age, technical batches), and unknown confounding $U$ using cate, using an optimal number of latent factors to

include, as suggested by cate:

$$y_j = \beta_{ij} x_i + \gamma M + \delta U \qquad (5)$$

Next, we performed the Bonferroni correction to the corresponding P-values $p_{ij}$ to identify genes associated with the genetic variant.

**Associating X-methylation with nearby gene expression**. To identify genes associated with DNA methylation of nearby CpGs (<250 Kb), we used a similar model as for *trans*-meQTL and *cis*-eQTL mapping. We associated methylation levels of CpGs $x_i$ with the observed expression values of a gene $y_j$ using a linear model, correcting for covariates M (i.e., cell counts, cohort, age, technical batches):

$$y_j = \beta_{ij} x_i + \gamma M \qquad (6)$$

The Bonferroni correction was used to determine significant CpG-gene pairs.

**Identifying genetic effects on autosomal DNA methylation**. To identify long-range effects (>5 Mb) of a genetic variant on DNA methylation at autosomal CpGs, we performed *trans*-meQTL mapping using all 3265 samples for which both genotype data and methylation data were available, as we expected these effects to be present in both women and men (Supplementary Fig. 6). For any genetic variant $i$ and CpG $j$, we fitted a linear model correcting for known covariates M (cell counts, cohort, age, technical batches), and unknown confounding U using *cate*:

$$y_j = \beta_{ij} x_i + \gamma M + \delta U \qquad (7)$$

The FDR was controlled within each set of corresponding P-values $p_{ij}$, to obtain a list of associated CpGs for a genetic variant $i$.

**Testing epistatic effects**. To test if the identified autosomal loci have any epistatic effects on X-chromosomal DNA methylation, we corrected the analysis for X-chromosomal *cis*-meQTLs. We first mapped *cis*-meQTLs (<250 Kb) on the X-chromosome by testing all nearby genetic variants for an effect on any of the X-chromosomal CpGs associated with one of the three autosomal loci. For any genetic variant $i$ and CpG $j$, we fitted a linear model correcting for known covariates M (cell counts, cohort, age, technical batches):

$$y_j = \beta_{ij}^X x_i + \gamma M \qquad (8)$$

We corrected for multiple testing using the Bonferroni procedure, selecting CpGs harboring *cis*-meQTLs. Next, we re-tested the effects of the autosomal loci on the X-chromosomal CpGs, but this time correcting for the strongest *cis*-SNP.

$$y_j = \beta_{ij}^X + \beta_{ij}^{auto} x_i + \gamma M \qquad (9)$$

**Annotations and enrichment tests**. CpGs were annotation using UCSC Genome Browser[68], histone marks and chromatin states data from the Blueprint Epigenome data[69], transcription factor binding site (TFBS) data from the Encode Project[70], and data on regions escaping X-inactivation[10]. All annotations were done based on the location of the CpG site using HG19/GRCh37.

The CGI track from the UCSC Genome Browser was used to map CpGs to CGIs. Shores were defined as the flanking 2 kb regions. All other regions were defined as non-CGI.

We obtained Epigenomics Roadmap ChIP-seq data on histone marks measured in blood-related cell types (the GM12878 lymphoblastoid cell line, the K562 leukemia cell line, and monocytes). We selected five different histone marks for which data measured in both men and women were available (H3K4me3, H3K4me1, H3K9me3, H3K27me3, H3K27ac). A CpG was said to overlap with any histone mark if it did so in any of the data sets.

We obtained Epigenomics Roadmap data on the 16 predicted core chromatin states data in blood-related cell types (the GM12878 lymphoblastoid cell line, the K562 leukemia cell line, and monocytes). A CpG was said to overlap with any chromatin state if it did so in any of the available data sets for that histone mark. Likewise, we obtained transcription factor binding data from the Encode Project, using blood-related cell types only (GM08714, GM10847, GM12878, GM12892, GM18505, GM18526, GM18951, GM19099, GM19193).

The degree of escape from X-inactivation for 632 TSSs has previously been established in 27 different tissues[10]. Within each tissue, each TSS was said to fully escape XCI, variably escape XCI, or be subject to XCI. We mapped each X-chromosomal CpG to the nearest such TSS, annotating each CpG with the accompanying scores for each of the 27 tissues. CpGs not in the vicinity of any such TSS (>10 kb, 4698 CpGs) were left unannotated.

In order to determine the enrichment of CpGs for any of the described genomic contexts, we used Fisher's exact test, where the used all X-chromosome CpGs as the background set.

**Code availability**. R code is available from https://git.lumc.nl/r.luijk/ChromosomeX. This repository describes the main analyses done.

### Data availability

Raw data were submitted to the European Genome-phenome Archive (EGA) under accession EGAS00001001077.

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

## Acknowledgements

This research was financially supported by several institutions: BBMRI-NL, a Research Infrastructure financed by the Dutch government (NWO, numbers 184.021.007 and 184.033.111); the UK Medical Research Council; Wellcome (www.wellcome.ac.uk; [grant number 102215/2/13/2 to ALSPAC]); the University of Bristol to ALSPAC; the UK Economic and Social Research Council (www.esrc.ac.uk; [ES/N000498/1] to CR); the UK Medical Research Council (www.mrc.ac.uk; grant numbers [MC_UU_12013/1, MC_UU_12013/2 to JLM, CR]); the Helmholtz Zentrum München – German Research Center for Environmental Health, which is funded by the German Federal Ministry of Education and Research (BMBF) and by the State of Bavaria; the Munich Center of Health Sciences (MC-Health), Ludwig-Maximilians-Universität, as part of LMUinno-vativ; the Wellcome Trust, Medical Research Council, European Union (EU), and the National Institute for Health Research (NIHR)- funded BioResource, Clinical Research Facility, and Biomedical Research Centre based at Guy's and St Thomas' NHS Foundation Trust in partnership with King's College London. Samples were contributed by LifeLines, the Leiden Longevity Study, the Netherlands Twin Registry (NTR), the Rotterdam Study, the Genetic Research in Isolated Populations program, the Cohort on Diabetes and Atherosclerosis Maastricht (CODAM) study, the Prospective ALS study Netherlands (PAN), Avon Longitudinal Study of Parents and Children (ALSPAC), International Schizophrenia Consortium, Cooperative Health Research in the Ausburg Region (KORA), TwinsUK. We thank the participants of all aforementioned biobanks and acknowledge the contributions of the investigators to this study. This work was carried out on the Dutch national e-infrastructure with the support of SURF Cooperative.

## Author contributions

Conceptualization, B.T.H., E.W.vZ., R.L., L.D., S.M.vdM.; Methodology, R.L., E.W.vZ.; Formal Analysis, R.L., H.W., C.K.W., E.H., E.C.M., J.L.M., P.M., M.M.N.; Resources, H.M., C.R., J.M., M.W., J.T.B., R.J., A.Z., L.F., P.tH., D.I.B., C.M.vD., M.vG., J.H.V., C.W., J.vM., P.E.S.; Writing – Original Draft, RL; Writing – Review & Editing, R.L., B.T.H.,

E.W.vZ., L.D., R.J., A.Z., L.F., P.tH., D.I.B., C.M.vD., M.vG., J.H.V., C.W., J.vM., P.E.S.; Visualization, R.L., B.T.H.; Supervision, B.T.H., E.W.vZ.

## Additional information

**Competing interests:** The authors declare no competing interests.

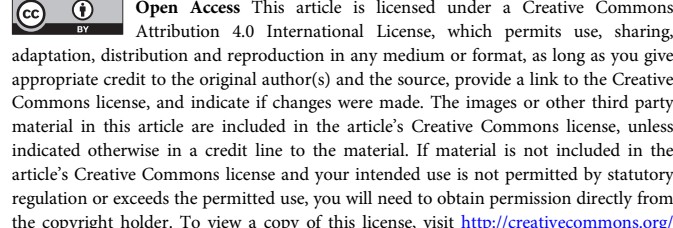

## BIOS Consortium

Marian Beekman[1], Ruud van der Breggen[1], Joris Deelen[1], Nico Lakenberg[1], Matthijs Moed[1], H. Eka D. Suchiman[1], Wibowo Arindrarto[2], Peter van't Hof[2], Marc Jan Bonder[15], Patrick Deelen[15], Ettje F. Tigchelaar[15], Alexandra Zhernakova[15], Dasha V. Zhernakova[15], Jenny van Dongen[16], Jouke J. Hottenga[16], René Pool[16], Aaron Isaacs[17], Bert A. Hofman[17], Mila Jhamai[18], Carla J.H. van der Kallen[19], Casper G. Schalkwijk[19], Coen D.A. Stehouwer[19], Leonard H. van den Berg[20], Michiel van Galen[2], Martijn Vermaat[2], Jeroen van Rooij[18], André G. Uitterlinden[18], Michael Verbiest[18], Marijn Verkerk[18], P. Szymon M. Kielbasa[21], Jan Bot[13], Irene Nooren[23], Freerk van Dijk[24], Morris A. Swertz[24] & Diana van Heemst[25]

[23]Present address: SURFsara, Amsterdam 1098 XG, The Netherlands. [24]Genomics Coordination Center, University Medical Center Groningen, University of Groningen, Groningen 9713 AV, The Netherlands. [25]Department of Gerontology and Geriatrics, Leiden University Medical Center, Leiden 2333 ZC, The Netherlands

