## [Peer Review File · Nature Communications]

Reviewer #1 (Remarks to the Author):

The paper by Luijk et. al. describes a very interesting approach driven by population genetics to examine the autosomal control of epigenetic effects during XCI. The authors show evidence that autosomal genetic variants (SNPs) are linked to / influence CpG island DNA-methylation of X-linked genes in the human. The analysis shows that such variants affect the methylation close to or within genes known to exhibit variable escape from X-chromosome inactivation. With this finding the paper adds a potentially important new aspect to the understanding of variable XCI in human.

The population based combined SNP/EWAS analysis start with a discovery cohort of females for which the authors analysed (used existing?) 450K DNA-methylation data and obtained (used existing? SNP data (not clear which)). They identify four autosomal sentinel SNPs (including one close to SMCHD1/METTL4, HBG2/TRIM6, ZSCAN9) showing a statistically significant (corrected P values) association with DNA-methylation changes and eQTLs in female but not in males. The claim that these loci are autosomal modifiers of XCI in human. The discovery of SMCHD confirms other previous genetic studies in the mouse.

Some remarks:

Overall the data integration and statistical analyses appear to be well conducted. However the documentation is poor and the authors avoid almost in the entire main text and the Methods to clearly outline "experimental" details for both the discovery and the validation.

It also remains unclear if any independent data validation was performed to support the epigenetic (sequencing) or genetic (functional testing) results.

The description in Mat&Meth on data processing are unclear it is not outlined how data were normalized, QC'ed and corrected for batch effects. This is particularly important given the small range of methylation changes responsible for the large p-value effects. The mean changes (information hidden in the figure legend) range between 1 and 1,6%. This is in the range of technical noise of the rather crude 450K array. Moreover this becomes even more of a problem when comparing data from many different cohorts/labs.

Are the four genetic loci also associated with methylation changes at other (autosomal) regions? The authors should consider to also test the potential influence of variants on (autosomal) genomic imprints and/or other autosomal loci (or even a whole chromosome as a control). If so, are these associations also exclusively found in females.

While the paper suggests a link of the autosomal variants to variable methylation the discussion on the functional link of the genes associated with the variants remains unclear.

Were the linkage-findings (reversely) checked for a cis role/influence of genetic variants located on the X chromosome? The autosomal variants could also have an epistatic effect on such X-chromosomal variants (which in turn have an impact on X-chr methylation)?

What are the allele frequencies for the identified allelic variants and are the effects observed based in small individual numbers carrying this allele?

Were the relevant 450k/SNP array probes checked for possible cross-hybridizing or unspecific effects, e.g. at pseudogenes, that were not listed by Chen et al 2013 (Epigenetics. 2013 Feb;8(2):203-9. doi: 10.4161/epi.23470. Epub 2013 Jan 11.)?

Overall the figure legends could be more informative.

Reviewer #2 (Remarks to the Author):

This is an interesting report of autosomal loci having an impact on level of X-chromosomal DNA methylation - particularly at genes that variably escape from X-chromosome inactivation (XCI). These results are novel and interesting; however, the manuscript was very challenging for me to follow. The paper itself is simply an abstract (referenced) and then text, rather than a more standard format. Further details and discussion are necessary to make the findings useful for the scientific community.

Figure 1: The results that are the focus of the manuscript are over-shadowed by larger other effects. Further exploration of these other effects would provide a basis for comparison of the effect of the female-limited effects.

For example, a table listing the genes potentially associated with these other peaks could also include the male / female p-values, genes impacted (perhaps top 5 or chromosomal location), and average level of impact on methylation. There seem to be some 'hits' that are similar between males and females, and some with substantially more impact on males - so an approach such as the use of

GO categories for the different 'hits' or 'target genes' might be informative. Additionally - is there a consistent enrichment for CpG islands over shores (which are often detected in genome-wide scans for differences, given their variability), shelves or open sea? And are most associated with promoters?

A clear table of such information (for both the female enriched and for all) would enhance the readability, and interpretation of the manuscript, and the usability of the results for the scientific community.

Given the substantial effect observed on the one CITED1 island, it would also be interesting to know if any of those other 'genes' (/peaks) impacted that same island.

Figure 2: Between PIN4 and CITED there is the escape gene RPS4X and the subject gene ERCC6L, which is not evident from the figure nor the text - why is PIN4 highlighted?

It would be helpful on the Circos plot to list the numbers of CpGs impacted at each site.

Panel B's legend is not complete. What is the colour scale? This comment would also apply to Figure 3, and even more to the supplementary figure where there is even less legend.

Figure 3:

HGB/TRIM6 - surely TRIM6 (or flanking genes) are the better candidate? Of note, TRIM6 was one of the XIST binding proteins identified by: Chu, C., Zhang, Q. C., da Rocha, S. T., Flynn, R. A., Bharadwaj, M., Calabrese, J. M., et al. (2015). Systematic discovery of Xist RNA binding proteins. *Cell*, 161(2), 404-416. <http://doi.org/10.1016/j.cell.2015.03.025>

Formatting comments as for Figure 2.

In the text, the total number of CpGs impacted was not mentioned. The data is there (277), but as the number is given for SMCHD1, it seems 'missing'.

In all cases, the strongest enrichment effect was in fact a depletion of effect on subject genes. Combined with the fact that escape genes lack DNA methylation at their islands, it is not clear to me that this effect is specific for variable escape genes. However, that comment does not detract from the intriguing observation that the effect seems to impact only a subset of genes.

Data analysis:

The manuscript methodology was also challenging for me to follow, with limited assistance from figure legends or discussion.

It was not apparent to me how the 'degree of escape' was obtained. The authors should explain the 5 point classification and how it is derived from the DNA methylation study cited.

Were mis-mapped/polymorphic/multiply aligning X probes removed from 450K data? As done previously (Price, M. E., Cotton, A. M., Lam, L. L., Farré, P., Emberly, E., Brown, C. J., et al. (2013). Additional annotation enhances potential for biologically-relevant analysis of the Illumina Infinium HumanMethylation450 BeadChip array. *Epigenetics & Chromatin*, 6(1), 4. <http://doi.org/10.1186/gb-2012-13-6-r44>).

What are the 'sentinal SNPs'? What impact do variants have on DNA methylation levels

ie - for different genotypes how much does methylation differ? Are these SNPs common enough to have homozygotes - in which case, what effect size is observed? Or were there relatively rare variants with larger effect (SMCHD1 seems to be a deletion)?

This information is critical to consider how much impact these variants have in the general population. Was the impact on the CITED1 island combinatorial (ie did having both SNPs increase the difference)?

Would it be possible to include the X chromosome (ie X-linked SNPs) in the analysis?

Supplementary data - in general the legends were limited (and non-existent for tables). For example, for S7 - what are bars, colours for XCI status, CpG, P value and effect size????

For tables:

In S2 what preferentially impacts male?

From S3 visual examination suggests that chromosome 10 is over-represented - does my eye deceive me?

I did not understand S6: The description "To link rs140837774 to a nearby gene we performed a cis-eQTL analysis using RNA-seq data from the females in the discovery set in our study (<250Kb, Table S6)" This seems to identify all genes (and multiple ZSCAN and ZKSCAN genes)

What is 'estimate' - column C, S7?

What are S8, S9 showing?

Reviewer #3 (Remarks to the Author):

I had the pleasure of reviewing the article from Luijk et al, which identifies I believe for the first time, 3 autosomal genetic variants that influence the extent of XCI in humans. Through this discovery, the study provides a nice confirmation of the role of SMCHD1 on human XCI, in addition to its well known role in mouse, and provides a further two candidate loci/genes previously unknown to be implicated in human XCI.

I found the statistical analyses used appropriate and for the most part well explained. Furthermore, three aspects of this study give me confidence in the results presented:

- the results are replicated in a meta-analysis of several independent cohorts;
- one of the 3 loci is in a very likely regulator of XCI;
- and there is supporting functional data linking the genetic variants to the expression of nearby genes and linking hypomethylation of X chromosome CpGs to the expression of nearby genes.

The effect sizes of the 3 variants found, a change of ~2 percentage points between a homozygous reference and an homozygous alternative individual, if I have understood it correctly, are very small. I would like to see this value put into context, what is the extent of variation in escape from XCI in whole blood, for example?

Until I found Sup. Table 5 it was unclear to me that the replication results come from the meta-analysis of replication done **separately** in the 6 cohorts. Please make this explicit in the methods in section "Replication of sentinel variants...".

The sentence in lines 113-114 "In particular, N⁶-methyladenosine is involved in XIST mediated XCI and key pluripotency factors." makes no sense.

In line 122 the overrepresentation of CPGs refers to which CpGs, all X-linked CpGs or only the SNP associated ones?

In line 134 it would be helpful to state what "degree of escape" concretely means, what sort of value is this?

In line 150 it would be helpful to add the words "The second of the three loci, sentinel SNP ..., " to clearly separate from the previous paragraphs about the 1st locus.

Was the third locus identified through the conditional analysis? It would be helpful to make this explicit in this paragraph (lines 174-183). Have you done any analysis that could implicate ZSCAN9 in similar pathways to SMCHD1/METTL4 therefore explaining how they independently regulate the same locus on the X chromosome?

In line 245 there is a type in "mased".

In line 247 add "[using base counts] over genes." Lines 248-250 are poorly explained, could you clarify what exactly was done? What is a "normal" gene?

Figures 2 and 3: state explicitly what red and blue (this one is missing) mean in the legend. A legend of the dot colours of the manhattan plot is missing. Add legend "X chromosome" to the appropriate segment of the circos plot.

In Figure 3, it looks like there are far more blue lines than the 6.5% hypermethylation associations mentioned in the legend. Perhaps the use of transparency for these lines would help, so that the blue lines do not stand out so much and the overlapping red lines are given more emphasis.

Make sure the references are proper, see reference 22.

Reviewer #1 (Remarks to the Author):

The paper by Luijk et. al. describes a very interesting approach driven by population genetics to examine the autosomal control of epigenetic effects during XCI. The authors show evidence that autosomal genetic variants (SNPs) are linked to / influence CpG island DNA-methylation of X-linked genes in the human. The analysis shows that such variants affect the methylation close to or within genes known to exhibit variable escape from X-chromosome inactivation. With this finding the paper adds a potentially important new aspect to the understanding of variable XCI in human.

The population based combined SNP/EWAS analysis start with a discovery cohort of females for which the authors analysed (used existing?) 450K DNA-methylation data and obtained (used existing? SNP data (not clear which)). They identify four autosomal sentinel SNPs (including one close to SMCHD1/METTL4, HBG2/TRIM6, ZSCAN9) showing a statistically significant (corrected P values) association with DNA-methylation changes and eQTLs in female but not in males. The claim that these loci are autosomal modifiers of XCI in human. The discovery of SMCHD confirms other previous genetic studies in the mouse.

We thank the reviewer for the thorough review, particularly regarding the data analysis. We feel these suggestions have allowed us to substantially improve the manuscript.

Some remarks:

Overall the data integration and statistical analyses appear to be well conducted. However the documentation is poor and the authors avoid almost in the entire main text and the Methods to clearly outline "experimental" details for both the discovery and the validation.

We have significantly expanded on the specific methods used in both the discovery and the validation set, by describing these in more detail using formal notation in the methods section.

It also remains unclear if any independent data validation was performed to support the epigenetic (sequencing) or genetic (functional testing) results.

In an attempt to experimentally reproduce the population-based analysis, we used shRNAs to knock down SMCHD1 and TRIM6 in the near diploid female RPE cell line (hTERT-RPE-1, ATCC, CRL-4000), and measured whether knock down affected mRNA levels of ALG13 and PIN4. While we did observe upregulation of ALG13 in some cases, the results were not always consistent (i.e. different shRNAs showed different effects). In addition, we assessed DNA methylation state of variably methylated regions of CITED1 and ALG13 in shSMCHD1 and shTRIM6 depleted RPE cells using Sanger bisulphite sequencing. Again, we obtained mixed results. While we could confirm variable X methylation of the candidate loci in RPE cells, we did not observe any significant loss of DNA methylation upon SMCHD1 or TRIM6 knock down. This could be explained by the fact that our population-based approach in a large sample has the statistical power to detect much smaller effect sizes than an *in vitro* experiment. In addition, this could be due to difficulties in recapitulating DNA methylation remodeling in cell models using shRNAs, or an inability to reverse XCI status using knock down approaches. More generally, we believe that the field currently lacks the tools to routinely and effectively evaluate findings from population epigenetic studies, which are characterized by subtle and long-term changes. This is in line with the complexities surrounding the functional follow-up of findings from genome-wide association studies,

despite the fact that such studies are being performed for over a decade now. So, in hindsight, we believe our attempt to reproduce our findings in vitro were perhaps naive.

The description in Mat&Meth on data processing are unclear it is not outlined how data were normalized, QC'ed and corrected for batch effects. This is particularly important given the small range of methylation changes responsible for the large p-value effects. The mean changes (information hidden in the figure legend) range between 1 and 1,6% . This is in the range of technical noise of the rather crude 450K array. Moreover this becomes even more of a problem when comparing data from many different cohorts/labs.

We have included a more detailed description of the methylation data processing. Furthermore, we now describe how we have corrected for batch effects in our analyses. Regarding the effect sizes, we now also mention the effect sizes in the main text, instead of only in the figure legends.

Are the four genetic loci also associated with methylation changes at other (autosomal) regions?

We have tested the effects of the identified loci on autosomal CpGs, identifying CpGs for the SMCHD1/METTL4 and ZSCAN9 loci only. We now include these results in the manuscript.

The authors should consider to also test the potential influence of variants on (autosomal) genomic imprints and/or other autosomal loci (or even a whole chromosome as a control). If so, are these associations also exclusively found in females.

We have tested for any effects of each of three loci on all available autosomal CpGs using meQTL mapping, thus including any available information on CpGs in imprinted regions. We have performed this analysis in female and male samples combined, as well as stratified by sex. The stratified results show very similar results (Figure S5). Any slight differences in test-statistics seem to be explained by the difference in sample sizes (1,398 male samples, 1,867 female samples). Any relevant results have been incorporated in the manuscript.

While the paper suggests a link of the autosomal variants to variable methylation the discussion on the functional link of the genes associated with the variants remains unclear.

In the paper, we discuss how we use eQTL mapping to link the identified variants to the nearby genes, which we now emphasize for each of the three loci. In addition, we report any known relations of the associated autosomal genes to XCI, as is the case for SMCHD1 and TRIM6. For ZSCAN9, we could not find a direct link with XCI.

Were the linkage-findings (reversely) checked for a cis role/influence of genetic variants located on the X chromosome? The autosomal variants could also have an epistatic effect on such X-chromosomal variants (which in turn have an impact on X-chr methylation)?

This is an interesting suggestion and now include an analysis testing if the main effect of an autosomal variant on an X-chromosomal CpG diminishes when a X-chromosomal cis-SNP is included in the model. The cis-SNP is selected to significantly affect the X-chromosomal CpG as well.

What are the allele frequencies for the identified allelic variants and are the effects observed based in small individual numbers carrying this allele?

We have included mean allele frequencies for the identified variants in the main text. The number of females per genotype were:

SMCHD1/METTL4: 358, 655, 308

TRIM6 /HBG2: 1308, 83, 0

ZSCAN9: 699, 875, 247

Were the relevant 450k/SNP array probes checked for possible cross-hybridizing or unspecific effects, e.g. at pseudogenes, that were not listed by Chen et al 2013 (Epigenetics. 2013 Feb;8(2):203-9. doi: 10.4161/epi.23470. Epub 2013 Jan 11.)?

In addition to removing any problematic probes identified by Chen et al (2013), we have checked the identified CpGs against the list provided by Price et al., and found only one overlapping probe, suggesting almost none of the identified CpGs here are non-specific.

Overall the figure legends could be more informative.

We have expanded upon the figure legends, making them more informative about each corresponding figure.

Reviewer #2 (Remarks to the Author):

This is an interesting report of autosomal loci having an impact on level of X-chromosomal DNA methylation - particularly at genes that variably escape from X-chromosome inactivation (XCI). These results are novel and interesting; however, the manuscript was very challenging for me to follow. The paper itself is simply an abstract (referenced) and then text, rather than a more standard format. Further details and discussion are necessary to make the findings useful for the scientific community.

We thank the reviewer for the helpful comments. We are particularly thankful for the detailed comments regarding the figures, which make the manuscript significantly easier to follow. In addition, the suggested paper on TRIM6 significantly strengthens the biological interpretation of our findings.

Figure 1: The results that are the focus of the manuscript are over-shadowed by larger other effects. Further exploration of these other effects would provide a basis for comparison of the effect of the female-limited effects.

For example, a table listing the genes potentially associated with these other peaks could also include the male / female p-values, genes impacted (perhaps top 5 or chromosomal location), and average level of impact on methylation. There seem to be some 'hits' that are similar between males and females, and some with substantially more impact on males - so an approach such as the use of GO categories for the different 'hits' or 'target genes' might be informative. Additionally - is there a consistent enrichment for CpG islands over shores (which are often detected in genome-wide scans for differences, given their variability), shelves or open sea? And are most associated with promoters?

A clear table of such information (for both the female enriched and for all) would enhance the readability, and interpretation of the manuscript, and the usability of the results for the scientific community.

We have significantly expanded on the interpretation of the 48 sentinel variants and their effects in both female and male samples, and included the results in Table S2. We have included more detailed information on the sentinel variants, and characterizations of the CpGs affected in male and female samples, including mean effect sizes and enrichments.

Given the substantial effect observed on the one CITED1 island, it would also be interesting to know if any of those other 'genes' (/peaks) impacted that same island.

We have looked up all the effects for the other genes/peaks, and found none of them had any effect on the same CpG island located near the CITED1 gene.

Figure 2: Between PIN4 and CITED there is the escape gene RPS4X and the subject gene ERCC6L, which is not evident from the figure nor the text - why is PIN4 highlighted? It would be helpful on the Circos plot to list the numbers of CpGs impacted at each site. Panel B's legend is not complete. What is the colour scale? This comment would also apply to Figure 3, and even more to the supplementary figure where there is even less legend.

We have expanded upon the figure legends for figures 2, 3, and the supplementary figure mentioned, describing in more detail what is shown.

In addition, we have included the other two genes in the figure. The reason PIN4 (and ALG13 elsewhere on the X-chromosome) is highlighted is that this gene is associated with changes in methylation of nearby genes. We now emphasize this in the figure legend.

Figure 3:

HGB/TRIM6 - surely TRIM6 (or flanking genes) are the better candidate? Of note, TRIM6 was one of the XIST binding proteins identified by: Chu, C., Zhang, Q. C., da Rocha, S. T., Flynn, R. A., Bharadwaj, M., Calabrese, J. M., et al. (2015). Systematic discovery of Xist RNA binding proteins. *Cell*, 161(2), 404-416. <http://doi.org/10.1016/j.cell.2015.03.025>
Formatting comments as for Figure 2.

The sentinel variant of this locus shows evidence for an eQTL effect on both genes. The stronger eQTL for HBG2 may be due to the fact that it has higher expression levels in our data (Table S6). The paper suggested by the reviewer, however, suggests a direct role of TRIM6 in XCI, making it a more likely candidate than HBG2. We have incorporated this reference in the manuscript and now propose TRIM6 as the most likely candidate to be involved in XCI. In addition, we also describe this figure in more detail.

In the text, the total number of CpGs impacted was not mentioned. The data is there (277), but as the number is given for SMCHD1, it seems 'missing'.

We now explicitly mention this number in the main text.

In all cases, the strongest enrichment effect was in fact a depletion of effect on subject genes. Combined with the fact that escape genes lack DNA methylation at their islands, it is not clear to me that this effect is specific for variable escape genes. However, that comment does not detract from the intriguing observation that the effect seems to impact only a subset of genes.

We concur with the observation of the reviewer. We speculate that methylation at fully inactivated CGIs is tightly regulated and may be impervious to the relatively mild functional effects of the autosomal variants we identified. To definitely answer the question raised by the reviewer, it may be of interest to analyze very large sample sets that are sufficiently powered to detect very small changes. We now note this in the manuscript.

Data analysis:

The manuscript methodology was also challenging for me to follow, with limited assistance from figure legends or discussion.

We have significantly expanded on the Methods section, including the addition of formal notation on the models used. Furthermore, we have expanded upon the figure legends, which now more clearly describe the accompanying figures.

It was not apparent to me how the 'degree of escape' was obtained. The authors should explain the 5 point classification and how it is derived from the DNA methylation study cited.

The addition of the 5-point scale may have been confusing when discussed next to the original 3-point scale, and does not necessarily add to the manuscript. Hence, we have chosen to only stick with the original 3-point scale determined in whole-blood provided by Cotton et al (*Hum. Mol. Genet.* **24**, 1528–1539 (2014)).

Were mis-mapped/polymorphic/multiply aligning X probes removed from 450K data? As done previously (Price, M. E., Cotton, A. M., Lam, L. L., Farré, P., Emberly, E., Brown, C. J., et al. (2013). Additional annotation enhances potential for biologically-relevant analysis of the Illumina Infinium HumanMethylation450 BeadChip array. *Epigenetics & Chromatin*, 6(1), 4. <http://doi.org/10.1186/gb-2012-13-6-r44>).

Ambiguously mapped probes were removed prior to analysis based on a report by Chen et al (see Methods, paragraph *DNA methylation data*, Discovery of cross-reactive probes and polymorphic CpGs in the Illumina Infinium HumanMethylation450 microarray. *Epigenetics* 8, 203–209 (2013)). In addition to this paper, we now have also checked the resource suggested by the reviewer (Price et al, 2013). Checking the identified CpGs in our data against the list provided by Price et al., we found only one overlapping probe, suggesting almost none of the identified CpGs here are non-specific.

What are the 'sentinal SNPs'? What impact do variants have on DNA methylation levels ie - for different genotypes how much does methylation differ? Are these SNPs common enough to have homozygotes - in which case, what effect size is observed? Or were there relatively rare variants with larger effect (SMCHD1 seems to be a deletion?)? This information is critical to consider how much impact these variants have in the general population. Was the impact on the CITED1 island combinatorial (ie did having both SNPs increase the difference)?

We have included a definition of the term “sentinel variant” in the main text (a term commonly used in the GWAS literature). In addition, we now include more details about these 48 variants in Table S2, including effect sizes for each of these variants, detailing if any variants have homozygotes.

The impact of the SMCHD1/METTL4 and ZSCAN9 loci on the CITED1 island seem to be independent. Correcting for each other does not change the test-statistics of either. We therefore conclude the effects are not combinatorial

Would it be possible to include the X chromosome (ie X-linked SNPs) in the analysis?

Reviewer 1 suggested to investigate if an autosomal SNP may have its effect on X-chromosomal methylation through X-chromosomal SNPs. We now include an analysis testing if the main effect of an autosomal variant on an X-chromosomal CpG diminishes when a X-chromosomal cis-SNP is included in the model. The cis-SNP is selected to significantly affect the X-chromosomal CpG as well.

Supplementary data - in general the legends were limited (and non-existent for tables). For example, for S7 - what are bars, colours for XCI status, CpG, P value and effect size????

We have significantly expanded on the figure legends, where we now describe in more detail what is shown in each figure.

For tables:

In S2 what preferentially impacts male?

In our analyses we have first identified 48 loci (with corresponding top SNPs/sentinel variants) in female samples. Only after we have identified these in females have we checked for similar effects in males. The overlap in CpGs affected per variant is generally high,

meaning there are no autosomal genetic variants with male-specific effects among these 48 loci.

From S3 visual examination suggests that chromosome 10 is over-represented - does my eye deceive me?

Table S3 shows autosomal genetic variants known to affect blood composition in two previous publications in independent samples (Orru, V. *et al*, *Cell* **155**, 242–256 (2013); Roederer, M. *et al.*, *Cell* **161**, 387–403 (2015)), and their effects on X-methylation in our data. The table and accompanying figure (Figure S4) show no significant effects of any of these variants on X-methylation.

I did not understand S6: The description "To link rs140837774 to a nearby gene we performed a cis-eQTL analysis using RNA-seq data from the females in the discovery set in our study (<250Kb, Table S6)" This seems to identify all genes (and multiple ZSCAN and ZKSCAN genes)

Indeed, multiple nearby genes are associated with this SNP, although ZSCAN9 is the strongest association. We edited the manuscript text to reflect this.

What is 'estimate' - column C, S7?

"estimate" refers to the associated regression coefficient. We have added legends for all tables, clarifying the column headers.

What are S8, S9 showing?

We have added legends for all tables, clarifying their contents. In this case, Table S8 shows the CpG-gene pairs that are significantly associated. Table S9 shows any significant meQTL effects of the 3 autosomal loci on methylation levels of autosomal CpGs.

Reviewer #3 (Remarks to the Author):

I had the pleasure of reviewing the article from Luijk et al, which identifies I believe for the first time, 3 autosomal genetic variants that influence the extent of XCI in humans. Through this discovery, the study provides a nice confirmation of the role of SMCHD1 on human XCI, in addition to its well known role in mouse, and provides a further two candidate loci/genes previously unknown to be implicated in human XCI.

I found the statistical analyses used appropriate and for the most part well explained. Furthermore, three aspects of this study give me confidence in the results presented:

- the results are replicated in a meta-analysis of several independent cohorts;
- one of the 3 loci is in a very likely regulator of XCI;
- and there is supporting functional data linking the genetic variants to the expression of nearby genes and linking hypomethylation of X chromosome CpGs to the expression of nearby genes.

The effect sizes of the 3 variants found, a change of ~2 percentage points between a homozygous reference and an homozygous alternative individual, if I have understood it correctly, are very small. I would like to see this value put into context, what is the extent of variation in escape from XCI in whole blood, for example?

We acknowledge that the identified effects are small. In compliance with the reviewer's suggestion, we now put these in perspective. Typically, inactivated regions show methylation values between 4% and 18% (Cotton et al, 2015)

Until I found Sup. Table 5 it was unclear to me that the replication results come from the meta-analysis of replication done *separately* in the 6 cohorts. Please make this explicit in the methods in section "Replication of sentinel variants...".

We have significantly expanded upon the methods section, also including more formal notation where appropriate. Here, we mention how the discovery and replication phases were conducted using which samples. We now describe the meta-analysis in the replication phase.

The sentence in lines 113-114 "In particular, N⁶-methyladenosine is involved in XIST mediated XCI and key pluripotency factors." makes no sense.

We agree that this sentence is unclear and We deleted the sentence in compliance with the reviewer's remark.

In line 122 the overrepresentation of CPGs refers to which CpGs, all X-linked CpGs or only the SNP associated ones?

The overrepresentation refers to an enrichment of the SNP associated CpGs, as compared to all X-linked CpGs, which we now specifically mention in the manuscript text.

In line 134 it would be helpful to state what "degree of escape" concretely means, what sort of value is this?

We noticed the different uses of the same annotation provided by Cotton et al may be confusing. Hence, we have removed this sentence from the manuscript.

In line 150 it would be helpful to add the words "The second of the three loci, sentinel SNP ..., " to clearly separate from the previous paragraphs about the 1st locus.

We have included the reviewer's suggestion in the manuscript, making it more obvious a new paragraph about a separate locus has started.

Was the third locus identified through the conditional analysis? It would be helpful to make this explicit in this paragraph (lines 174-183). Have you done any analysis that could implicate ZSCAN9 in similar pathways to SMCHD1/METTL4 therefore explaining how they independently regulate the same locus on the X chromosome?

These loci were indeed identified using conditional analyses, which we now explicitly state in the manuscript text.

As the number identified ZSCAN9 effects is limited to this one CpG island, we could not link ZSCAN9 to either SMCHD1/METTL4.

In line 245 there is a typo in "mased".

We have corrected this typo, changing it into "masked".

In line 247 add "[using base counts] over genes." Lines 248-250 are poorly explained, could you clarify what exactly was done? What is a "normal" gene?

We have edited this paragraph to make it clearer. In addition, we have removed the term "normal gene".

Figures 2 and 3: state explicitly what red and blue (this one is missing) mean in the legend. A legend of the dot colours of the manhattan plot is missing. Add legend "X chromosome" to the appropriate segment of the circos plot.

We have significantly expanded on all figure legends, where we now also include descriptions of the meaning of colors in the figures. In addition, each Circos plot now includes the text "X chromosome" (figures 2, 3, s7).

In Figure 3, it looks like there are far more blue lines than the 6.5% hypermethylation associations mentioned in the legend. Perhaps the use of transparency for these lines would help, so that the blue lines do not stand out so much and the overlapping red lines are given more emphasis.

We have adjusted the transparency of the blue lines, which does indeed help a bit. However, many red lines concentrate around several locations, making it seem like there aren't as many red lines as the figure description says.

Make sure the references are proper, see reference 22.

We have fixed the reference mentioned.

Reviewer #1 (Remarks to the Author):

The authors addressed all critical comments of my previous review adequately. The paper has significantly improved in terms of clarity and reproducibility. As said before the paper adds a couple of important new aspects to the understanding of human X-inactivation. This really elegant and elaborate story should be published as presented.

Reviewer #2 (Remarks to the Author):

Upon revision, the manuscript is now much clearer; however, the effect size is quite small, and I believe needs further clarification.

In the abstract the term 'hypomethylation' is used; which is accurate, but conjures images of a substantially greater effect than 1%. I would suggest using a modifier such as slight hypomethylation?

Given the small effect, it seems critical to examine it further. While the biological relevance of the autosomal associations is highly suggestive of a real effect, validation is still lacking. The failure of the shRNA experiments is difficult to interpret without seeing the data. Furthermore, I remain unclear on what exactly is the impact on DNA methylation.

SMCHD1/METTL4 variant – mean effect size per allele of 1% - does this mean average loss at 56 CpGs of 1%? What is range (of loss) and what is the range of methylation AT THESE 56 CpGs?

Presentation of the assessment of the DNA methylation state for the shRNA experiment might address this question.

Further, I did not understand the statement "typical methylation levels for inactivated genes are between 4 and 18%". Genes subject to inactivation show up to 80% methylation at CpGs at island-containing promoters. The number presented is presumably the total female methylation – but still seems far too low – should it not be up to 40-50%???

Given agreement of the authors to the previous comment that variable escape genes 'have methylation to lose', I believe it would be informative to see what happens if the effect is monitored as % pre-existing methylation.

Since these are variably escaping genes, is the small difference at all samples or a larger change at a few sites? Perhaps a larger effect at a small number of individuals explains the lack of validation by shRNA?

Further supporting a link with variable expression, an association between methylation and expression is observed – and the reader is referred to supplemental methods.

This seems important data that should be presented in the main text of the body. Presumably this is based on larger variations than the 1% observed??

Again, for TRIM/HGB 'mean effect size of 1.6% per allele' – what is meant by allele??? Per CpG?

If per CpG then what is the pre-existing levels – a plot (violin, or box & whisker) of the DNAm across females for the 57 SMCHD1 associated CpGs would be informative (harder for 257 TRIM/HGB). Or perhaps demonstrate for the 17 CpGs that overlap between SMCH and ZSCAN.

Finally, the discussion surrounding the association on X not clear to me.

Minor comments:

Paper switches back and forth from 3-4 sentinel variants.

The introduction has {ref}

Reviewer #3 (Remarks to the Author):

The presentation of the methodology, which was previously hard to follow, has improved in this version and the new analyses introduced are appropriate and relevant. I have a few further remarks:

Did the author test for variants on X regulating XCI in "trans" (same chromosome but $\geq 250\text{Kb}$)? They have done a trans analysis for autosomal SNPs, so why only in cis for the X chromosome SNPs?

Results, paragraph 1: I would have it mentioned in the main text that the cell counts, cohort, age, etc were included as covariates in the association test.

Supplementary Fig 3: What exactly is being tested here and how? What are the "cell count" variables tested? How were the counts obtained in the discovery cohorts, were they all obtained using the same platform?

Results, paragraph 5: "typically methylation levels for inactivated regions are between 4% and 18%" -> What exactly is meant by "inactivated" regions? The relevant statistic here is to report the variation in extent of XCI escape. I think the numbers are correct, just not properly conveyed.

Results, paragraph 5: "The ... variant was associated with altered methylation levels of 57 ... CpGs ... with hypomethylation at 56 of those CpGs (98.2%, binomial $P = 8.5 \times 10^{-13}$)" -> This is an enrichment of hypomethylation compared to what (same question for the same test for the 2nd and 3rd variants as well)? The percentage 98.2% is unnecessary when the numbers are clearly 56 out of 57.

Results, paragraph 9: Is the T allele associated with lower expression of both HBG2 and TRIM6 ("The sentinel variant was associated with decreased HBG2 expression ... and also with decreased expression of the nearby gene TRIM6") or are they in opposite directions ("The high-HBG2 and low-TRIM6 expression allele (T allele) was associated with...")?

Methods, Annotations -> which blood-related cell types were used exactly?

Minor comments:

Abstract: "All three loci were consistently associated with hypomethylation of CpG islands" - this sentence is imprecise, what is meant by "consistently associated"? Also, by definition, one allele confers higher methylation while the other confers lower methylation.

Results, paragraph 1, last sentence: Shouldn't this reference Table 8 rather than Table 3?

Introduction, paragraph 3: "{ref}" -> fix missing reference

Results, paragraph 1: $P < 5 \times 10^{-8}$ is the threshold used for females, whereas $P < 1.1 \times 10^{-6}$ is the largest p-value in males that is under the 0.05 threshold (according to the Methods). So either report the largest p-value below the thresholds for both males and females, or the threshold used (females $P < 5 \times 10^{-8}$, males $P \leq 0.05$).

Results paragraph 1: "autosomal three loci" -> "three autosomal loci"

Results, paragraph 2: "were caused due to" -> were due to

Methods, Discovery cohorts: In addition, sex, age, cell counts, and were obtained -> fix sentence

Throughout the text (optional suggestion): why the nomenclature X-linked CpGs and X-linked genes, why not X-chromosome genes and X-chromosome CpGs? Using the words "associated" and "linked" in the same sentence makes it a little confusing at times, also sometimes you use the first formulation and sometimes the second.

Results, paragraph 7: "Supplementary table 98" -> reference correct table

Results, paragraph 9: "The sentinel variant was associated with decreased HBG2 expression" -> The alternative allele of the sentinel variant was associated with decreased HBG2 expression.

Results, last paragraph: the sentinel SNP was significantly and consistently associated -> what does consistently mean?

Results, last paragraph: "all 19 CpGs show consistently effects for both loci" -> again, what does this mean? direction and magnitude or just direction?

Results, last paragraph: "X-chromosome CpGs associated with genetic variation ... " -> Break sentence into two.

Discussion, first sentence: "ZSCAN9), and are all associated with" -> ZSCAN9), all of which are associated with.

Methods, Identifying female specific effects: "We repeated this process until no new independent effects were identified" -> how many times?

Methods, Replication of sentinel variants: "discussed in 68):" -> remove extra ":".

Reviewer #1 (Remarks to the Author):

The authors addressed all critical comments of my previous review adequately. The paper has significantly improved in terms of clarity and reproducibility. As said before the paper adds a couple of important new aspects to the understanding of human X-inactivation. This really elegant and elaborate story should be published as presented.

Reviewer #2 (Remarks to the Author):

Upon revision, the manuscript is now much clearer; however, the effect size is quite small, and I believe needs further clarification.

In the abstract the term 'hypomethylation' is used; which is accurate, but conjures images of a substantially greater effect than 1%. I would suggest using a modifier such as slight hypomethylation?

We agree a change in wording may better reflect the effect sizes. Therefore, we have changed the wording to 'mild hypomethylation'.

Given the small effect, it seems critical to examine it further. While the biological relevance of the autosomal associations is highly suggestive of a real effect, validation is still lacking. The failure of the shRNA experiments is difficult to interpret without seeing the data. Furthermore, I remain unclear on what exactly is the impact on DNA methylation.

We concur that the effect size is not large. However, this is inherent to any approach focusing on common genetic variation like genome-wide association studies. Natural selection will prevent variants with large effects to become common. To nevertheless detect such effects in a robust manner, large sample sizes, stringent significance thresholds and replication in independent cohorts are the gold standard. We adhered to this gold standard of population genomics.

That said, we do agree with the reviewer that the addition of mechanistic studies would take the interest in the manuscript to a significantly higher level. That is why we tried whether we could set up a relatively straight-forward experimental system in vitro to assay genes for an effect of genes on female X methylation. As described in our previous rebuttal, we were unable to do so. Importantly, our experimental approach was unable to reproduce the effect of SMCHD1, even though this is well-established. One reason could be that our experiments would only show effects on XCI maintenance, whereas the population genomics approach could also show effects on XCI initiation. Indeed, a recent study from Neil Brockdorff's lab reported that a Caspr/Cas9-mediated *Smchd1* knock-out in wild type mouse embryonic fibroblasts did not affect gene expression patterns on the X-chromosome (<https://doi.org/10.1101/342147>). This is consistent with our observations that shRNA-mediated of SMCHD1 knock down is not sufficient to re-activate genes on the X-chromosome.

We strongly believe that mechanistic studies of female X methylation, particularly at variably escaping regions (characterized by variable and relatively low methylation levels), require meticulous, large-scale and lengthy experiments that need to be developed from scratch for two scenarios, namely genes affecting establishment and maintenance of X methylation. We were not unwilling to put in extra effort in additional experiments, and we did. However, the

realization that complex experiments are required to make them meaningful, meant to us that they are beyond the scope of the current manuscript. This does not deter from the robust evidence from our population genomics approach.

In compliance with the reviewer's criticism and to acknowledge that mechanistic studies will be required to provide definite proof for an involvement in variable X methylation of the genes we identified, we now discuss this limitation of our study in the Discussion section.

SMCHD1/METTL4 variant – mean effect size per allele of 1% - does this mean average loss at 56 CpGs of 1%? What is range (of loss) and what is the range of methylation AT THESE 56 CpGs?

The mean effect size does indeed refer to the average loss at these 56 CpGs. Specifically, it refers to the average loss per reference allele, so between the homozygotes (AA) and heterozygotes (Aa). This means that the difference between homozygotes (AA and aa) would be twice this loss – or 2%. To get a better idea of the distribution of effect sizes and the distribution of methylation for those CpGs, we have included a supplementary figure showing these distributions for each of the loci (suppl fig 5).

Presentation of the assessment of the DNA methylation state for the shRNA experiment might address this question.

Please, refer to answer to earlier point.

Further, I did not understand the statement “typical methylation levels for inactivated genes are between 4 and 18%”. Genes subject to inactivation show up to 80% methylation at CpGs at island-containing promoters. The number presented is presumably the total female methylation – but still seems far too low – should it not be up to 40-50%???

Thank you for spotting this inaccuracy. We mistakenly wrote “inactivated” where we meant “escaping”. Genes escaping inactivation show low levels of DNA methylation. We corrected the error.

Given agreement of the authors to the previous comment that variable escape genes ‘have methylation to lose’, I believe it would be informative to see what happens if the effect is monitored as % pre-existing methylation.

That is an interesting suggestion. Since the average methylation level of variably escaping CpGs is ~23%, this would mean that the relative changes in methylation as a percentage of this 17% would be ~4% (so the average difference in DNA methylation between the two homozygote genotypes would change from an absolute 2% to a relative ~8%). However, for easier comparison across CpGs (with different mean methylation levels), we feel it will be clearer to most readers if we adhere to absolute differences even though the effects sizes appear smaller.

Since these are variably escaping genes, is the small difference at all samples or a larger change at a few sites? Perhaps a larger effect at a small number of individuals explains the lack of validation by shRNA?

We show the scatter plots of the top associated CpGs for the three loci discussed in the manuscript. They do not seem to show a small number of individuals causing the associations. For completeness' sake, we list the number of samples per genotype below.

Number of samples per genotype:

TRIM6 /HGB2: 1631, 228, 8

SMCHD1/METTL4: 482, 929, 456

ZSCAN9: 713, 896, 258

Further supporting a link with variable expression, an association between methylation and expression is observed – and the reader is referred to supplemental methods.

This seems important data that should be presented in the main text of the body.

Presumably this is based on larger variations than the 1% observed??

We have added a section in the main methods section, and include this table as a main table. We test all CpGs associated with any of the sentinel variants for an association with the expression of nearby genes. In doing this, as suggested by the reviewer, we use the full variation observed for each CpG.

Again, for TRIM/HGB ‘mean effect size of 1.6% per allele’ – what is meant by allele??? Per CpG?

If per CpG then what is the pre-existing levels – a plot (violin, or box & whisker) of the DNAm across females for the 57 SMCHD1 associated CpGs would be informative (harder for 257 TRIM/HGB). Or perhaps demonstrate for the 17 CpGs that overlap between SMCH and ZSCAN.

Similar to the question for the SMCHD1/METTL4 locus, the mean effect size refers to the average loss at these CpGs for every extra reference allele. Here, this too implies the difference between homozygotes would be twice this loss – or 3.2%. We refer to the same supplementary figure as mentioned above (sup fig 5).

Finally, the discussion surrounding the association on X not clear to me.

We assume the reviewer is referring to testing for any epistatic effects. Previously, we assumed the autosomal loci have their effects on X-chromosomal DNA methylation through cis-regulation of autosomal genes (uncovered using cis-eQTL mapping). One of the reviewers suggested the effects could also be mediated by X-chromosomal genetic variation. We therefore tested this by correcting for X-chromosomal genetic variants in the neighbourhood of the identified X-chromosomal CpGs.

Minor comments:

Paper switches back and forth from 3-4 sentinel variants.

We have changed the discussion of the different sentinel variants such that this is minimized. The change between 3 and 4 sentinel variants is due to the fact that we initially discovered 4 loci, but 1 of those loci did not replicate. Only for the 3 replicated loci do we report on follow-up analyses.

The introduction has {ref}

We have fixed this, and added the corresponding reference.

Reviewer #3 (Remarks to the Author):

The presentation of the methodology, which was previously hard to follow, has improved in this version and the new analyses introduced are appropriate and relevant. I have a few further remarks:

We very much appreciate the time the reviewer has taken to carefully read of our manuscript and to spot final typos and ambiguities.

Did the author test for variants on X regulating XCI in "trans" (same chromosome but \geq 250Kb)? They have done a trans analysis for autosomal SNPs, so why only in cis for the X chromosome SNPs?

The initial analysis focuses on autosomal genetic effects on X-methylation, as previous studies using mice suggests such effects do exist. Hence, we did not use genetic variation on X itself.

As suggested by another reviewer (reviewer 1) in commenting on a previous version of the manuscript, we tested the hypothesis that these autosomal genetic loci may have effects that are amplified by X-chromosomal genetic variation in cis. We therefore tested for an epistatic effect of these autosomal loci, thereby only taking into account X-chromosomal genetic variants near any of the X-chromosomal CpGs identified earlier to be associated with these autosomal genetic loci.

Our study may suggest additional research questions like the identification of long-range effects on X methylation of X genetic variants as proposed by the reviewer. However, we think such analyses will be difficult to interpret in the context of epistatic effects and may clutter the manuscript.

Results, paragraph 1: I would have it mentioned in the main text that the cell counts, cohort, age, etc were included as covariates in the association test.

We agree that for the sake of clarity and transparency this is desirable. Hence, we have added these in the main text.

Supplementary Fig 3: What exactly is being tested here and how? What are the "cell count" variables tested? How were the counts obtained in the discovery cohorts, were they all obtained using the same platform?

The analysis described in Suppl Fig 3 tests the effects of all 4 sentinel variants initially identified to have female-specific effects on X-chromosomal methylation with all of the cell count measures in our data. These include lymphocytes, neutrophils, monocytes, eosinophils, basophils, and red blood cell counts, and are all obtained using the same platform. The analysis is corrected for all the other covariates that we have in our data, this includes age and known batches.

We have clarified this in the description of the figure.

Results, paragraph 5: "typically methylation levels for inactivated regions are between 4% and 18%" -> What exactly is meant by "inactivated" regions? The relevant statistic here is to report the variation in extent of XCI escape. I think the numbers are correct, just not properly conveyed.

This is an inaccuracy that we now fix. We mistakenly wrote "inactivated" where we meant "escaping". We have changed this in the text, so that it now makes more sense: genes escaping inactivation show low levels of DNA methylation.

Results, paragraph 5: "The ... variant was associated with altered methylation levels of 57 ... CpGs ... with hypomethylation at 56 of those CpGs (98.2%, binomial $P = 8.5 \times 10^{-13}$)" -> This is an enrichment of hypomethylation compared to what (same question for the same test for the 2nd and 3rd variants as well)? The percentage 98.2% is unnecessary when the numbers are clearly 56 out of 57.

The null hypothesis used in this enrichment states that the percentage of positive/negative effects should be 50%. Hence, we also explicitly state the percentage of observed positive/negative associations, as this is relevant for the binomial test performed.

Results, paragraph 9: Is the T allele associated with lower expression of both HBG2 and TRIM6 ("The sentinel variant was associated with decreased HBG2 expression ... and also with decreased expression of the nearby gene TRIM6") or are they in opposite directions ("The high-HBG2 and low-TRIM6 expression allele (T allele) was associated with...")?

We have changed the wording to make it clearer the reference allele (T allele) is associated with decreased expression of both genes (HBG2, TRIM6).

Methods, Annotations -> which blood-related cell types were used exactly?

In the analyses, we have used the GM12878 lymphoblastoid cell line, the K562 leukemia cell line, and monocytes. We now include this in the methods section

The available cell counts were lymphocytes, neutrophils, monocytes, eosinophils, basophils, and red blood cell counts. We now explicitly mention these in the first paragraph of the methods section.

Minor comments:

Abstract: "All three loci were consistently associated with hypomethylation of CpG islands" - this sentence is imprecise, what is meant by "consistently associated"? Also, by definition, one allele confers higher methylation while the other confers lower methylation.

We have tried to convey the point that the effects of the loci are skewed towards either hypo-, or hypermethylating their target CpGs. We have changed the wording to reflect this better.

Results, paragraph 1, last sentence: Shouldn't this reference Table 8 rather than Table 3?

The reference should indeed be to Supplementary Table 3, where the test statistics reflect the associations with the X-chromosomal CpGs corrected for any nearby genetic variants, if they are available in our data.

Supplementary Table 8 shows the main effects

Introduction, paragraph 3: "{ref}" -> fix missing reference

We noticed the error, and now fix this by including the right reference.

Results, paragraph 1: $P < 5 \times 10^{-8}$ is the threshold used for females, whereas $P < 1.1 \times 10^{-6}$ is the largest p-value in males that is under the 0.05 threshold (according to the Methods). So either report the largest p-value below the thresholds for both males and females, or the threshold used (females $P < 5 \times 10^{-8}$, males $P \leq 0.05$).

Results paragraph 1: "autosomal three loci" -> "three autosomal loci"

Results, paragraph 2: "were caused due to" -> were due to

We have corrected the above two typos.

Methods, Discovery cohorts: In addition, sex, age, cell counts, and were obtained -> fix sentence

We have added the technical batch information, fixing the sentence.

Throughout the text (optional suggestion): why the nomenclature X-linked CpGs and X-linked genes, why not X-chromosome genes and X-chromosome CpGs? Using the words "associated" and "linked" in the same sentence makes it a little confusing at times, also sometimes you use the first formulation and sometimes the second.

We agree with the reviewer's point. To avoid the confusion, we now use the reviewer's suggestion of X-chromosome, instead of X-linked.

Results, paragraph 7: "Supplementary table 98" -> reference correct table

We now refer to the right table.

Results, paragraph 9: "The sentinel variant was associated with decreased HBG2 expression"
-> The alternative allele of the sentinel variant was associated with decreased HBG2 expression.

For clarity, we have adopted the reviewer's suggestion.

Results, last paragraph: the sentinel SNP was significantly and consistently associated -> what does consistently mean?

We agree the term "consistently" is a bit vague. Hence, we now refrain from the use of this word. What we meant to say is that the reference allele of this variant is associated with hypomethylation at all CpGs. As we also explicitly mention this later in the paragraph, we now removed this term.

Results, last paragraph: "all 19 CpGs show consistently effects for both loci" -> again, what does this mean? direction and magnitude or just direction?

What is meant is the direction of the effects, which is now made explicit.

Results, last paragraph: "X-chromosome CpGs associated with genetic variation ... " -> Break sentence into two.

The message of this sentence indeed requires two sentences. We have edited the text according to the reviewer's suggestion.

Discussion, first sentence: "ZSCAN9), and are all associated with" -> ZSCAN9), all of which are associated with.

We have adopted the reviewer's suggestion, making the sentence run more smoothly.

Methods, Identifying female specific effects: "We repeated this process until no new independent effects were identified" -> how many times?

We have done this genome-wide analysis 49 times: the first analysis yields all individual genetic variants associated with X-chromosomal DNA methylation. Next, re-doing the entire analysis while correcting for the top SNP was repeated until no more statistically significant variants were obtained, which was after 47 of such iterations. This led to a total of 48 sentinel variants. We now explicitly mention this in the manuscript.

Methods, Replication of sentinel variants: "discussed in 68):" -> remove extra ":".

We have replaced the ":" with a comma ","

Reviewer #2 (Remarks to the Author):

I thank the authors for their detailed responses and highlighted manuscript. My concerns have been met.

Reviewer #3 (Remarks to the Author):

I am satisfied that the authors have addressed the comments of my previous review adequately.